# Development of 4-[4-(Anilinomethyl)-3-phenyl-pyrazol-1-yl] Benzoic Acid Derivatives as Potent Anti-Staphylococci and Anti-Enterococci Agents

**DOI:** 10.3390/antibiotics11070939

**Published:** 2022-07-13

**Authors:** Hansa Raj KC, David F. Gilmore, Mohammad A. Alam

**Affiliations:** 1Department of Chemistry and Physics, The College of Sciences and Mathematics, Arkansas State University, Jonesboro, AR 72401, USA; hansa.kc@smail.astate.edu; 2Department of Biological Sciences, The College of Sciences and Mathematics, Arkansas State University, Jonesboro, AR 72401, USA; dgilmore@astate.edu

**Keywords:** pyrazole, persisters, *Staphylococcus aureus*, *Enterococcus faecalis*, resistance

## Abstract

From a library of compounds, 11 hit antibacterial agents have been identified as potent anti-Gram-positive bacterial agents. These pyrazole derivatives are active against two groups of pathogens, staphylococci and enterococci, with minimum inhibitory concentration (MIC) values as low as 0.78 μg/mL. These potent compounds showed bactericidal action, and some were effective at inhibiting and eradicating *Staphylococcus aureus* and *Enterococcus faecalis* biofilms. Real-time biofilm inhibition by the potent compounds was studied, by using Bioscreen C. These lead compounds were also very potent against *S. aureus* persisters as compared to controls, gentamycin and vancomycin. In multiple passage studies, bacteria developed little resistance to these compounds (no more than 2 × MIC). The plausible mode of action of the lead compounds is the permeabilization of the cell membrane determined by flow cytometry and protein leakage assays. With the detailed antimicrobial studies, both in planktonic and biofilm contexts, some of these potent compounds have the potential for further antimicrobial drug development.

## 1. Introduction

Antimicrobial resistance (AMR) poses one of the greatest threats to human health around the world and it has been argued that AMR could kill 10 million people per year by 2050 [1]. The six pathogens, SPEAKS (*Staphylococcus aureus*, *Pseudomonas aeruginosa*, *Escherichia coli*, *Acinetobacter baumannii*, *Klebsiella pneumoniae*, and *Streptococcus pneumoniae*) are the leading causes of AMR deaths. *S. aureus* and its methicillin-resistant variant (MRSA) caused more than 100,000 deaths worldwide in 2019. On the other hand, ESKAPE pathogens (*Enterococcus faecium*, *S. aureus*, *K. pneumoniae*, *A. baumannii*, *P. aeruginosa*, and *Enterobacter* spp.) cause the majority of nosocomial infections and these bacteria have the ability to evade the existing treatments [2]. *S. aureus* infection is a major problem in the United States, which causes almost 100,000 infections and 20,000 deaths every year [3]. *S. aureus* infections are caused by different strains, including methicillin-sensitive *S. aureus* (MSSA), MRSA, vancomycin-intermediate *S. aureus* (VISA), and vancomycin-resistant *S. aureus* (VRSA). While most staphylococcal infections are due to MRSA, any *S. aureus* infection can be dangerous and lethal [4]. Hospitals are the major places for highly drug-resistant pathogens, such as MRSA, increasing the menace of hospitalization kills instead of cures [5,6,7,8]. MRSA has emerged as a multidrug-resistant pathogen in nosocomial infections and this pathogen is bypassing HIV in terms of fatality rate [9].

*Enterococcus faecium*, a Gram-positive bacterium, is a normal microbiotum of the gastrointestinal tract (GI) and female genital tracts of humans and most other animals, including insects. This bacterium can cause a variety of problems when introduced into other parts of the body. Endocarditis and bacteremia are the most serious and life-threatening diseases caused by *E. faecium* [10]. *E. faecium* readily acquires antibiotic resistance genes and has become 80% vancomycin and 90% ampicillin-resistant [11,12]. About 30% of all healthcare-associated enterococcal infections are caused by vancomycin-resistant (VRE) strains, and these resistant strains are increasingly becoming resistant to other antibiotics. In 2017, VRE caused 54,500 patient hospitalizations and 5400 deaths in the United States. According to the CDC’s National Healthcare Safety Network, solid organ transplant units reported vancomycin-resistant *E. faecium* as the most common cause of central line-associated bloodstream infections [13]. Furthermore, enterococci biofilms contribute to 25% of all catheter-associated urinary tract infections [14].

Pyrazole nucleus has widely been found in approved drugs, such as apixaban (Eliquis^@^), celecoxib (Celecox^@^), and several others (https://go.drugbank.com/categories/DBCAT000650 accessed on 8 July 2022). A myriad number of synthetic derivatives of this azole have been reported for their a nticancer [15,16], antibacterial [17], antiviral [18], and several other therapeutic properties [19]. In our research on azole derivatives as potent antibacterial [20,21,22] and antineoplastic agents [23,24,25], we found pyrazole-derived hydrazones are potent growth inhibitors of *A. baumannii* [21,26] and the aniline derivatives of pyrazoles are potent growth inhibitors of Gram-positive bacteria [22,27]. In this article, we report data on the antimicrobial activities, and the modes of action, of a series of hit compounds (Figure 1) [28,29]. The featured compounds (MIC ≤ 32 μg/mL) were selected for further studies following screening for antimicrobial activity, low toxicity against human embryonic kidney cell lines (HEK293) and their low lipophilicity, compared to other compounds [28,29].

## 2. Results and Discussion

### 2.1. MIC Studies

The identified hit compounds (**1**–**11**) were tested against 13 Gram-positive bacterial strains (Table 1). The mono-substituted compounds were moderate growth inhibitors of these bacterial strains. 3-Fluorophenyl aniline (**1**) is a weak inhibitor of tested bacterial strains. The 3-chlorophenyl derivative (**2**) showed better activity than the fluoro derivative (**1**), with MIC values in the range of 50 to 6.25 μg/mL. 3-Bromo (**3**) and 3-trifluoromethyl (**4**) derivatives showed similar antimicrobial properties. Overall, disubstituted compounds, except the 4-fluoro-3-methyl substituted derivative (**5**), showed better activity than those of the mono-substituted compounds (**1**–**4**). The 4-fluoro-3-methyl aniline derivative (**5**) showed moderate growth inhibition activity, with MIC values as low as 12.5 μg/mL against *Bacillus subtilis*. The 3-chloro-4-methyl derivative (**6**) was a potent growth inhibitor of the tested strains. This compound (**6**) inhibited the growth of *S. aureus* strains, with MIC values in the range of 3.12–6.25 µg/mL. This compound was also potent against enterococci, *B. subtilis*, and *Staphylococcus epidermidis* bacteria. The 4-bromo-3-methyl aniline derivative (**7**) showed similar activities as its chloro analogue (**6**). Bis(trifluoromethyl)aniline (**8**) was the best compound in the series, with potent activity across the tested strains. This disubstituted aniline derivative (8) was very active in inhibiting the growth of *S. aureus* strains, with MIC values as low as 0.78 µg/mL. *Enterococcus faecalis* and *E. faecium* strains were inhibited effectively, with an MIC value of 3.12 µg/mL, by this compound (**8**). *B. subtilis* and *S. epidermidis* strains were inhibited, with MIC values 0.78 and 6.25 µg/mL, respectively. 4-Fluoro-3-trifluoromethyl-substituted derivative (**9**) showed varied activity against different strains, with MIC values as low as 3.12 μg/mL against *S. aureus* Newman and *B. subtilis* strains. Chloro and bromo analogues (**10** and **11**) were very potent across the *S. aureus* strains, with an MIC value of 3.12 μg/mL. These compounds were also effective against the enterococci and *B. subtilis* strains.

### 2.2. Bactericidal Properties and Time-Kill Assay

The minimum bactericidal concentration (MBC) is the minimum concentration that kills > 99.9% of a bacterial species within 18–24 h of treatment. If the MBC value of an antibacterial agent is not more than four times the MIC value, then the antibacterial agent is considered to be bactericidal. If the MBC/MIC value is more than four, then the antibacterial agent is considered to be bacteriostatic. Generally, the bacteriostatic agents inhibit protein synthesis and the bactericidal agents target the cell wall of bacteria [30]. As Table 2 shows, the hit compounds (**6**–**10**) were bactericidal for the *B. subtilis* strain, except for compound **11**, which is a bacteriostatic agent. These compounds were bactericidal for the *S. epidermidis* strain. Compound **6** was bactericidal or bacteriostatic, based on the strain of *S. aureus* bacteria. Compounds **7**–**11** were bactericidal for the tested *S. aureus* strains. The bactericidal action was consistent with the compounds being membrane disruptors. We performed the time kill assay at 4 × MIC concentration to determine the bactericidal properties of our hit compounds. As can be seen in Figure 2, potent compounds **8**, **10**, and **11** killed the bacteria within 6 h. Compound **6** did not kill the bacterial cells even after 24 h. Vancomycin eliminated bacteria up to the level of the detection limit within 8 h. These observations corresponded to the bacteriostatic nature of compound **6** and bactericidal properties of compounds **8**, **10** and **11**. These observations agreed with the MBC values (Table 2) of the compounds and the positive control, vancomycin. Since bactericidal activity was determined from bacteria exposed to compounds for 20 h, it cannot be completely ruled out that low colony counts could have arisen from a failure of a subpopulation to recover from a viable non-culturable state brought on by antibiotic exposure [31].

### 2.3. Activity against Biofilms

We studied the biofilm inhibition properties of three potent compounds against the *S. aureus* ATCC 25923 strain (Figure 3). Compound (**8**) was very effective at inhibiting the growth of *S. aureus* biofilm at 2 × MIC concentration but potency decreased at lower concentrations. Compounds **10** and **11** showed very strong biofilm inhibition properties at all the tested concentrations: 2×, 1×, and 0.5 × MIC values. The positive control, vancomycin (Van), showed good activity at 2×, and 1 × MIC values, but this approved drug inhibited only ~60% biofilm formation at 0.5 × MIC treatment. We studied the biofilm eradication properties of these potent compounds. Among these compounds, the bis(trifluoromethyl)aniline derivative (**8**) was a moderate biofilm eradicator of *S. aureus*, which eradicated ~80% biofilm at 2 × MIC. Decreasing the concentration of this compound reduced the biofilm elimination property significantly. Compound **10** eliminated the *S. aureus* biofilm effectively in all three concentrations. Compound **11** showed good biofilm eradication properties at 2× and 1 × MIC values, albeit only ~55% biofilm eradication at 0.5 × MIC value. The positive control, vancomycin, showed weak biofilm elimination activity as compared to our lead compounds **10** and **11**. The ability of compounds **10** and **11** to significantly inhibit and eradicate biofilm at the concentration of 0.5 × MIC was particularly noteworthy, indicating that the observation was not just a side effect of the bacteriostatic or bactericidal properties of the compounds.

We also studied the biofilm inhibition properties of the molecules against *E. faecalis* ATCC 29212. Compound **8** inhibited >90% biofilm growth in all three tested concentrations. Compound **10** inhibited >90% biofilm formation at 2 × MIC value, but the inhibition potency decreased gradually at lower concentrations. The third potent lead compound (**11**) inhibited the biofilm formation of *E. faecalis*, at lower potency. Vancomycin showed similar potency at 2× and 1 × MIC values but, nevertheless, biofilm inhibition potency decreased significantly at the sub-MIC value. In the *E. faecalis* biofilm eradication studies of our hit compounds, we observed potent activity comparable to the positive control. In particular, compound **10** again showed moderate biofilm inhibition and eradication below the MIC and MBC concentrations.

We determined the minimal biofilm eradication concentration (MBEC), the lowest concentration of compound sufficient to prevent growth from a treated biofilm, to further determine the potency of the effective lead compounds. As shown in Table 3, compounds **8, 10**, and **11** eradicated the established biofilms of *S. aureus* ATTCC 25923 and *E. faecalis* ATCC 29212 at concentrations 2× and 4 × MIC, respectively. Compounds were less effective at eradicating the biofilm of *S. aureus* USA300, with concentrations as high as 50 µg/mL (16 × MIC) needed to reach the MBEC (compound **10**). Our studies showed that the positive control used in MIC tests, vancomycin, was not effective in eradicating the tested biofilms. These studies are very significant, as *S. aureus* biofilms are very challenging to treat with the existing antibiotics [32]. Similarly, *E. faecalis* readily forms biofilms, which are recalcitrant to existing treatments [33].

### 2.4. Real-Time Monitoring of S. aureus Biofilm

Bacteria within a biofilm matrix are better protected from host defenses and antibiotics. Regular doses of antibiotics can reduce the biofilm but rarely eliminate it [34,35]. We treated biofilms produced by two strains of *S. aureus* with the lead compounds at various concentrations, as well as with conventional antibiotics, and monitored real-time biofilm growth/inhibition using the Bioscreen C Pro. This test depends on the production of planktonic cells from the biofilm or any increase in the biofilm itself. The effects of various concentrations of compounds **8**, **11**, vancomycin and daptomycin on *S. aureus* ATCC 25923 and *S. aureus* USA300 biofilm, as compared to the DMSO treated negative control, are shown in Figure 4 and Figure 5. The biofilm minimum inhibitory concentration (BMIC) is defined as the lowest concentration of the compound at which the optical density remains at baseline at the 24 h time point. The BMIC for compound **8** against *S. aureus* ATCC 25923, as measured kinetically using Bioscreen C Pro, was observed to be 4 × MIC of planktonic cells (6.25 µg/mL) (Figure 4A) and 4 × MIC (12.5 µg/mL) against *S. aureus* USA300 (Figure 4A). BMIC for compound **11** was 2 × MIC (6.25 µg/mL) for *S. aureus* ATCC 25923 and 4 × MIC for *S. aureus* USA300, while MPC against both the strains was 4 × MIC concentrations (Figure 4B and 5B). BMIC for vancomycin was observed to be 8 × MIC and 16 × MIC for *S. aureus* ATCC 25923 and *S. aureus* USA300, respectively. Daptomycin exhibited a BMIC value to be 16 × MIC, and an MPC value being >16 × MIC, against *S. aureus* ATCC 25923 biofilm (Figure 4D).

### 2.5. Activity against Persisters

Persisters are dormant phenotypic variants of bacteria that are recalcitrant to killing by antibiotics. Persisters are a chronic and continuous nidus of infection that can result in treatment failure [36]. Bacteria comparable to persisters can be produced by allowing a culture to grow until the stationary phase is well established. We tested our potent lead molecules (**8**, **10**, and **11**) for their ability to eliminate such persister cells of *S. aureus* ATCC 700699 (Figure 6). Cells were incubated in phosphate buffered saline (PBS), conditions, under which they do not resume growth. As can be seen (Figure 6A), our compounds (**8**, **10**, and **11**) reduced the persisters’ viability after 4 h at 8 × MIC treatment. The positive controls, approved antibiotics (gentamicin and vancomycin), did not show any effectiveness against the persisters and showed similar viability as the negative control (2.5% DMSO). The lack of bactericidal activity by vancomycin is to be expected, since cell wall inhibiting antibiotics lack effectiveness against non-growing cells. To observe the persister elimination pattern with time, persister cells were incubated in PBS with various concentrations of compounds (**8**, **10** and **11**). Aliquots were drawn out every 2 h for a viable colony count for up to 8 h total time (Figure 6B–D). Compound **8** considerably reduced the concentration of persisters at 2×, 4×, and 8 × MIC values. *S. aureus* persisters were markedly reduced by 16 × MIC treatment. Compound **10** was more effective in decreasing persisters at similar concentrations. This compound (**10**) decreased *S. aureus* persisters up to the detection limit at 8 × MIC treatment in 8 h. The 16 × MIC treatment eliminated the persisters in 6 h. Although the hit compound **11** showed weak activity at 2 × MIC treatment, it showed comparable potency at higher treatment doses. It was found that 8 × MIC and 16 × MIC treatments eliminated persisters within 8 h and 6 h, respectively. Based on the anti-persister activity, we could conclude that our potent compounds were very effective at killing *S. aureus* persisters in vitro. This finding warrants further development of these compounds as antibiotics.

### 2.6. Multistep Resistance Studies

The success of antibiotics over the years has been threatened by the evolution of antimicrobial resistance (AMR). Microbial pathogens have the ability to avoid or delay death upon exposure to antibiotics that were supposed to kill them [27,28]. We studied the ability of *S. aureus* and *E. faecalis* strains to develop resistance against our potent lead compounds (Figure 7). *S. aureus* ATCC 700699 was treated with lead compound **8** and the bacteria became slightly less susceptible by the third day (two-fold increase in MIC). However, no further resistance against this compound (**8**) was seen up to 14 days in our study. Lead compound **11** at 1 × MIC was effective against *S. aureus* up to 7 days, after which the MIC increased only to two-fold of the original. Bacteria developed resistance to vancomycin to two-fold on the sixth passage and four-fold after the ninth day of treatment. *E. faecalis* did not develop resistance to our compounds through all 14 days of the study, while the MIC of vancomycin increased two-fold on the 7th day. Based on these observations, *S. aureus* failed to develop resistance to our compounds easily, and *E. faecalis* did not develop resistance at all up to 14 days. While these studies show a failure to acquire resistance by mutations, they do not rule out the possibility of gaining resistance by horizontal gene transfer in natural environments if such resistance exists.

### 2.7. Membrane Permeability Studies

Determination of the mode of action of antimicrobial compounds is crucial for the development of new drugs, as well as for new therapeutic applications for existing drugs [29]. To examine whether membrane permeabilization has a role in the mode of action, kinetic fluorescence measurement and flow cytometry analysis were performed in the presence of the fluorescent dye propidium iodide (PI) (Figure 8). PI fails to accumulate in cells with healthy membranes, but if the bacterium is unable to exclude this red fluorescent dye, it binds to DNA by intercalation increasing its fluorescence. If the test compound disrupts the permeability of the bacterial membrane, PI enters the cell and can be detected fluorometrically once it binds to DNA.

For kinetic fluorescence measurement, *S. aureus* ATCC 700699 was incubated with the test compounds at various MIC concentrations in PBS containing 10 µg/mL Propidium Iodide (PI) at 37 °C in 96-well black microtiter plate. Figure 8A–C show kinetic fluorescence measurements of PI after treatment of bacterial cells with various MIC concentrations of the compounds. After the addition of the compound at 5 min (red arrow indicates compound addition), fluorescence was observed to increase rapidly at higher MIC concentrations for both test compounds **8** and **11**, while there was a slight increase in fluorescence at lower MIC values. For compound **8**, there was a rapid increase in fluorescence at 8 × MIC until 33 min, after which the fluorescence intensity was above the detectable limit (>100,000 RFU) for the device. At 4 × MIC, which is the MBC for this compound, there was a rapid increase in fluorescence up to the 9th minute, after which the fluorescence remained constant till the end of the experiment (Figure 8A). For compound **11**, there was a rapid increase in fluorescence at 8× and 4 × MIC concentrations up to the 13th and 29th minutes, after which fluorescence was above the detectable limit. A rapid increase was observed at 2 × MIC (below the MBC) up to the 13th minute, with a slight decrease followed by constant fluorescence till the end (Figure 8B). No increase in fluorescence was observed up to 8 × MIC concentration of vancomycin, indicating no membrane permeabilization for the tested highest concentration for this conventional drug.

Figure 8D shows a protein leakage assay that was performed to determine the leakage of cellular proteins. Once the bacterial cell membrane has been irreversibly disrupted, cellular contents, such as DNA, RNA, and proteins, will leak [30,31]. An increase in protein concentration with increasing MIC doses, in comparison to negative controls, untreated and DMSO treated samples, were observed, indicating the possibility of cell membrane damage and protein leakage.

Flow cytometry analysis was performed to determine the membrane-disrupting ability of the potent compounds. Various MIC concentrations of compounds **8** and **11** were treated with *S. aureus* ATCC 700699, and then stained with PI to measure PI permeabilization into the bacterium cells. As observed in Figure 9A, only 7.09% of the cells were stained in 1% DMSO treated cells, which was our negative control. Almost 96.11% of cells were stained in the positive control sample that was treated with 70% ethanol (Figure 9B). Vancomycin, which was used as technical control, showed cell staining similar to DMSO treated cells, indicating non-permeabilization of bacterial cells at 4 × MIC concentration (Figure 9C). Compound **8** treated cells showed higher PI intensity, with a good dose-response correlation. Treatment with 1×, 2×, and 4 × MIC concentrations showed 10.18%, 12.94%, and 67.82% PI-permeant cells, respectively (Figure 9D–F). Compound **11** demonstrated better PI permeability with dose-response in comparison to our lead compound **8.** For compound **11**, PI permeability was 12.63% and 34.93%, respectively, at 1 × MIC and 2 × MIC. Compound **11** at 4 × MIC treatment showed 91.23% PI-permeant cells. These results demonstrated that the compounds might have directly damaged or interfered with the membrane functions, as shown by the failure to exclude PI. However, it cannot be ruled out that membrane destabilization was a result of cell death, and not the direct cause.

## 3. Materials and Methods

### 3.1. Antimicrobial Compounds *(**1–11**)*

The compounds (**1–11**) were synthesized as reported by us previously [28,29]. The purity of the compounds was determined by ^1^H NMR before testing.

### 3.2. Minimum Inhibitory Concentration (MIC)

MIC for the compounds was determined using the standard microdilution technique recommended by the Clinical and Laboratory Standards Institute (CLSI). The starting concentration of the compounds was 32 µg/mL and a 2-fold dilution was performed along the 96-honeycomb well plate column to determine the MIC. MIC was confirmed in at least two occurrences of three replicates performed. A concentration was considered inhibitory when no visually detectable turbidity was present after about 20 h. Slow growth (up to 2 log increase over 20 h) could still fail to produce turbidity and would be considered inhibited.

### 3.3. Minimum Bactericidal Concentration (MBC)

MBC was determined for some compounds against various bacterial strains under study. After determination of MIC in 96-well columns, non-turbid well contents, including the MIC wells, were diluted 10-fold (10^0^, 10^1^ and 10^2^) and then spot plated on TSA plates to quantify viable cells. The plates were incubated and colonies were counted to determine the percentage of viable cells, compared to the initial CFU/mL. The MBC was defined as the lowest concentration that reduced bacterial concentration by at least 99.9%.

### 3.4. Time-Kill Assay

Time-kill assay was performed following the methodology described earlier [22]. The exponential phase bacterial culture of *S. aureus* ATCC 700699 was diluted to ~5 × 10^6^ CFU/mL in CAMHB, then treated with 4 × MIC of the test compounds and incubated at 35 °C. Every 2 h, the treated aliquot was diluted 10-fold in PBS and viable cells were quantified by viable colony count on TSA plates.

### 3.5. Biofilm Inhibition and Eradication Assays

Biofilm inhibition and eradication assays were performed as described previously [32]. In brief for biofilm inhibition assay, overnight bacterial culture was suspended to 0.5 McFarland standard in sterile PBS, which was then diluted 1:1000 in cation adjusted Muller Hilton Broth (CAMHB) supplemented with 1% glucose. Bacterial suspension, along with desired concentrations of the compound, were placed in triplicates into 96-well flat-bottom plates and incubated for 24 h at 35 °C. After incubation, each well was washed with PBS thrice to remove planktonic cells, then dried in the oven at 60 °C for 15 min. The wells were stained with crystal violet (0.1% *w*/*v*) for 15 min, then washed with deionized water to remove unstained dyes and dried in the oven for 15 min. Acetic acid (33%) was added to the stained wells to solubilize crystal violet, then optical density was measured using a Biotek Cytation 5 plate reader at 620 nm excitation wavelength.

For destruction assay, bacterial biofilm was first established in 96-well plates by culturing the overnight bacteria in CAMHB supplemented with 1% glucose and incubating 24 h at 35 °C. After incubation, the wells were washed thrice by PBS, then CAMHB, containing various concentrations of the compound, were added in triplicates to challenge the established biofilm in the wells. After an additional 24 h of incubation, washing, drying, staining by crystal violet, solubilizing stained biofilm, and measuring optical density were performed as described above. Percentage biofilm inhibition and destruction were calculated using the optical density of DMSO treated wells as negative control and media-only wells as a positive control.

### 3.6. Antibiofilm Studies Using Calgary Device

Calgary biofilm device was utilized to determine the minimum biofilm eradication concentration (MBEC) following the methodology described earlier [27]. Briefly, biofilm was established on the lid pegs of the device using overnight bacterial culture diluted in CAMHB supplemented with 1% glucose at 35 °C for 24 h. After incubation, the lid was removed, washed with PBS in a fresh 96-well plate, then transferred to the “challenge plate” which contained 2-fold serial diluted compounds in PBS. The challenge plate was incubated for 24 h, then the lid was transferred to the next 96-well plate containing fresh CAMHB with 1% glucose and further incubated for 24 h. Following the final incubation, the plates were observed for visible turbidity to determine MBEC, which was the lowest concentration of compound well that resulted in no turbidity. Each compound was tested in triplicates to confirm the MBEC value.

### 3.7. Real-Time Monitoring of S. aureus Biofilm

Real-time monitoring of biofilm growth/inhibition was performed following the methodology described by Elkhatib et al. [34] with few modifications. Overnight culture of *S. aureus* was diluted (1 × 10^6^ CFU/mL) in CAMHB with 1% glucose and seeded (150 µL) into 100-well polystyrene honeycomb plates to establish biofilm. Several control wells received only culture medium and no cells so that no biofilm would be produced. The plate was incubated in the chamber of Bioscreen C Pro (Growth curves USA, Piscataway, NJ, USA) for 24 h at 37 °C without shaking. Following incubation, each well of the plate was washed thrice with sterile PBS to remove planktonic cells. CAMHB with 1% glucose, containing desired concentrations of the compound, were prepared and transferred to the biofilm established wells (200 µL). Control wells consisted of DMSO treated and wells without bacteria (medium only). The prepared honeycomb plate was placed in the preheated chamber of Bioscreen C Pro programmed to maintain the temperature of 37 °C for 96 h without shaking. A wide-band filter with a spectrum range of 400–600 nm was programmed to measure the optical densities every 1 h up to 96 h of real-time monitoring of biofilm growth/inhibition.

### 3.8. Persister Cell Killing Assay

Persister cell killing assay was performed following the methodology described earlier [22]. Briefly, *S. aureus* was grown in CAMHB for 24 h by shaking at 200 rpm at 35 °C to the stationary phase. The stationary phase cells were washed thrice in PBS and then diluted to around 10^8^ CFU/mL in the same buffer. The diluted persister suspension was treated with the desired concentration of the compound in a sterile 10 × 75 mm plastic culture tube by shaking at 200 rpm at 35 °C. At a desired interval of time, aliquots were taken in microcentrifuge tubes, washed with PBS twice, 10-fold serially diluted, and spot plated in tryptic soy agar (TSA) plates for viable colony count to determine CFU/mL for each treatment.

### 3.9. Multi-Step Resistance Assay

The ability of *S. aureus* ATCC 700699 (MRSA) and *E. faecalis* ATCC 29212 to develop resistance to the test compounds was investigated by multi-step resistance assay. The minimum inhibitory concentration (MIC) was determined on the first day, which was the first passage of the assay. Sub-MIC well from the first passage 96-well plate was used as bacterial inoculum to determine MIC for second passage. The experiment was repeated similarly for further passages and MIC for each passage was recorded for up to 14 passages to investigate resistance development.

### 3.10. Kinetic Fluorescence Measurements to Detect Membrane Permeabilization Using Propidium Iodide

The kinetic fluorescence measurement to detect membrane permeabilization was conducted following the methodology described by Boix-Lemonche et. al. [35] with few modifications. Exponential phase *S. aureus* ATCC 700699 (MRSA) was harvested by centrifugation, washed and diluted in PBS (1 × 10^8^ CFU/mL). Propidium iodide (PI) was added to a final concentration of 10 µg/mL, followed by incubation for 15 min in the dark. After incubation, the mixture was vortexed and 195 µL was transferred to wells of a black 96-well plate. The plate was placed in the chamber of the Biotek Cytation^TM^ 5 plate reader at 37 °C and fluorescence was measured at excitation and emission wavelengths 535 nm and 617 nm, respectively, every minute for 5 min, or until readings were stabilized. After this, the plate was ejected and compounds at various concentrations (5 µL dissolved in DMSO) were added to pre-designated wells. The plate was further monitored with fluorescence reading parameters mentioned earlier every minute up to a total of 60 min, shaking continuously.

### 3.11. Protein Leakage Assay

Protein leakage from the bacterial cells, due to damage to the bacterial cell membrane, was determined using the methodology described by Xie et al. [36], with a few modifications. Exponential phase bacterial cells were collected by centrifugation at 5000× g for 10 min, followed by washing thrice with PBS and suspending in the same buffer. The bacterial suspension was treated with various MIC concentrations of compound **11** at 37 °C by shaking at 200 rpm for 2 h. The treatment sample was then centrifuged at 10,000× g for 5 min and the supernatant was used to estimate the protein concentration, using a standard Bradford assay. Untreated bacterial cell samples and DMSO-treated samples, which were negative control for this experiment, were also processed accordingly.

### 3.12. Flow Cytometry for Membrane Permeability

Membrane permeabilization using propidium iodide (PI) was further quantified using flow cytometry analysis as described previously [37]. Briefly, exponential phase *S. aureus* ATCC 700699 cells grown in CAMHB were harvested at 4000 rpm for 10 min, followed by washing twice with PBS, and then diluting to ~10^5^ CFU/mL in the same buffer. The bacterial suspension was incubated with various desired MIC concentrations of compounds for 30 min at 35 °C, while shaking at 200 rpm. The cells were again harvested from the treatment by centrifugation, and washed with PBS to remove excess compounds. The washed cells were incubated for 30 min with 10 µg/mL of PI in PBS at 4 °C in the dark. The cells were then washed again to remove unbound dye, and, then, data were recorded with an excitation wavelength of 488 nm (Phycoerythrin-Texas Red A filter), using BD FASAria^TM^ cell sorter (BD Biosciences, Franklin Lakes, NJ, USA). After that, 1% DMSO and 70% ethanol-treated cells were taken as the negative and positive controls, respectively. Conventional antibiotic vancomycin was used as technical control.

## 4. Conclusions

We studied the antimicrobial properties of 11 lead compounds and four of these compounds were potent growth inhibitors of different Gram-positive bacterial strains, with some MIC values at sub-µg/mL concentration against several of the tested strains. These potent compounds were very effective against bacteria, both in planktonic and biofilms contexts. Two potent compounds (**8** and **11**) were very potent biofilm inhibitors at 2 × MIC doses, and were consistently better than the positive control, vancomycin. In the real-time effect of potent compounds on biofilm inhibition studies, hit compounds were several-fold more effective than the control antibiotics, vancomycin and daptomycin. *E. faecalis* bacteria failed to develop any resistance against two of our compounds over 14 days. The most potent compounds were bactericidal and directly, or indirectly, caused membrane damage, as shown by protein leakage assays and propidium iodide permeability assays using flow cytometry.

## Figures and Tables

**Figure 1 antibiotics-11-00939-f001:**
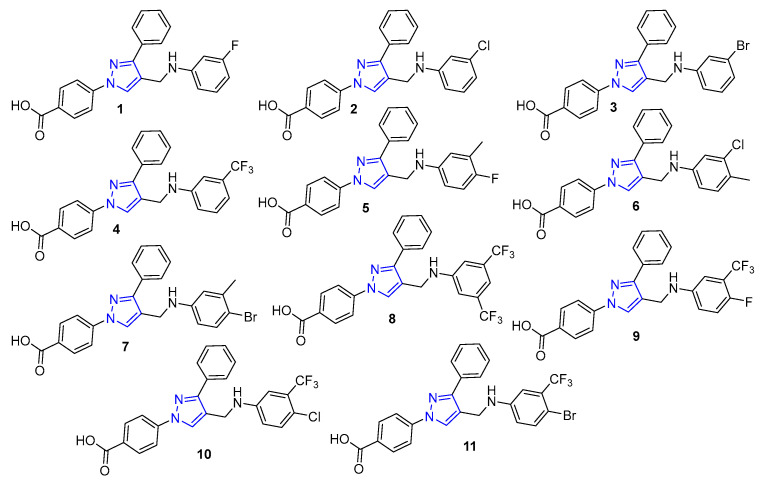
Structure of the hit compounds.

**Figure 2 antibiotics-11-00939-f002:**
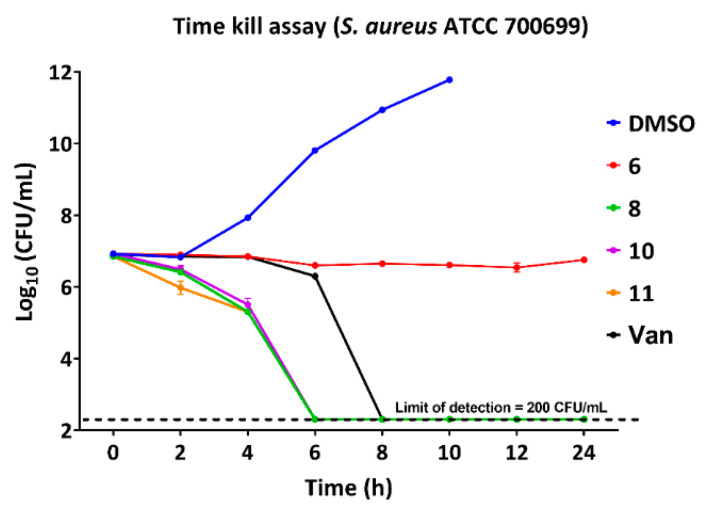
Time-kill assay (TKA) of *S. aureus* ATCC 700699 against the potent compounds treated at 4 × MIC concentration. Each data point represents the mean value of viable colony counts performed in triplicate and the error bars represent standard deviation values.

**Figure 3 antibiotics-11-00939-f003:**
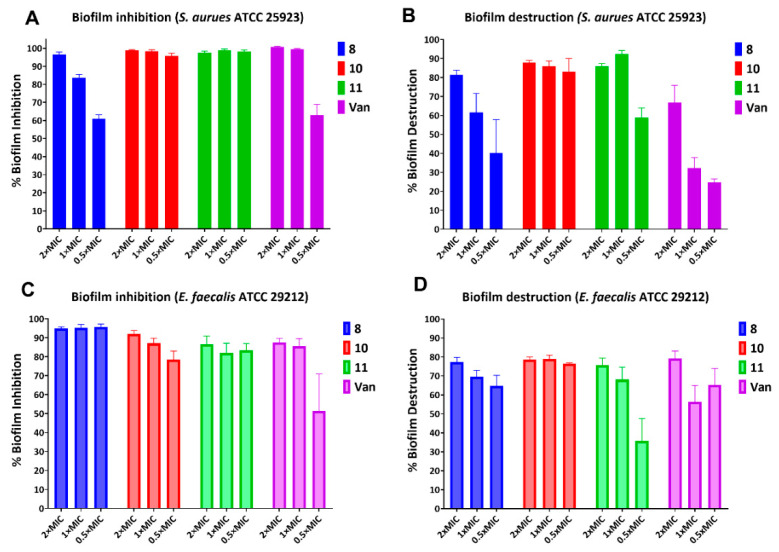
Biofilm inhibition of *S. aureus* (**A**) and *E. faecalis* (**C**) and biofilm destruction of *S. aureus* (**B**) and *E. faecalis* (**D**) by potent compounds (**8**, **10**, and **11**). Vancomycin (Van) is the positive control. Data based on retention of crystal violet by biofilm as compared to DMSO (the solvent of compound diluent as negative control). Error bars represent standard deviation values of the readings obtained in triplicates.

**Figure 4 antibiotics-11-00939-f004:**
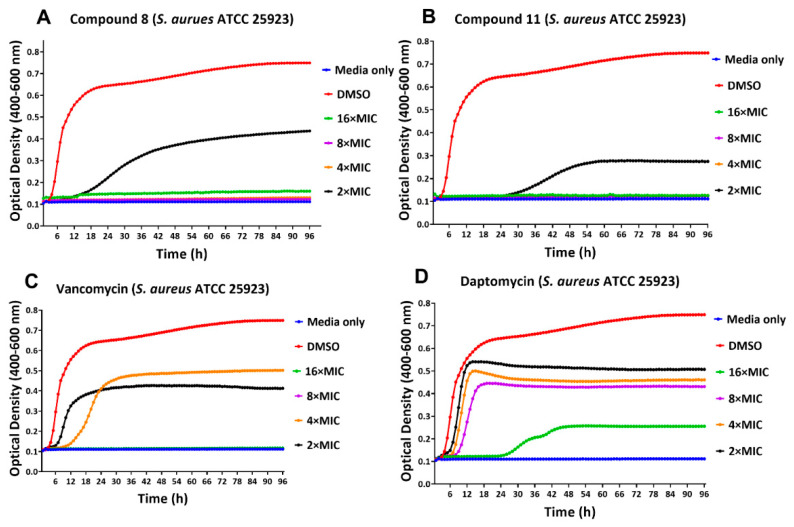
Real-Time monitoring of *S. aureus* ATCC 25923 biofilm. Effect of various concentrations of compound **8** (**A**), compound **11** (**B**), Vancomycin (**C**) and Daptomycin (**D**) against *S. aureus* ATCC 25923 biofilm growth, as monitored by Bioscreen C Pro up to 96 h. Legends indicate treatment of preformed biofilm with DMSO control and various MIC concentrations of each compound. An additional control consisted of growth medium with no pre-formed biofilm in the wells. Each data point represents the mean value of the optical density readings performed in triplicates.

**Figure 5 antibiotics-11-00939-f005:**
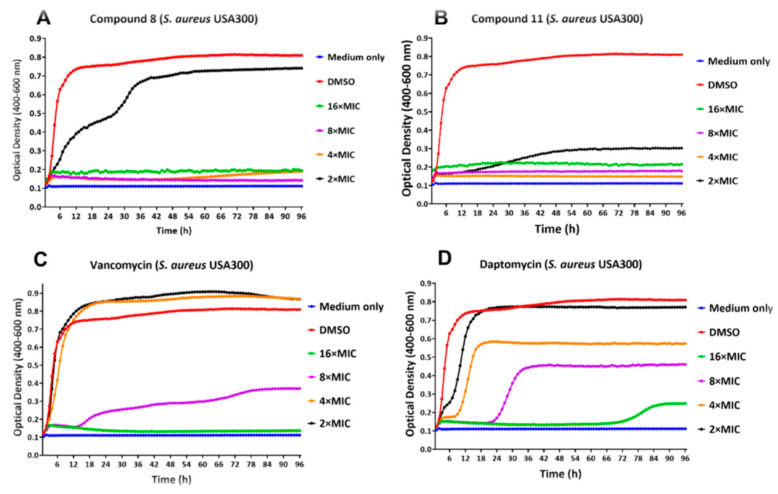
Real-time monitoring of *S. aureus* USA300 biofilms. Effect of various concentrations of compound **8** (**A**), compound **11** (**B**), Vancomycin (**C**) and Daptomycin (**D**) against *S. aureus* USA300 biofilm growth, as monitored by Bioscreen C Pro up to 96 h. Legends indicate treatment of preformed biofilm with DMSO control and various MIC concentrations of each compound. An additional control consisted of growth medium with no pre-formed biofilm in the wells. Each data point represents the mean value of the optical density readings performed in triplicates.

**Figure 6 antibiotics-11-00939-f006:**
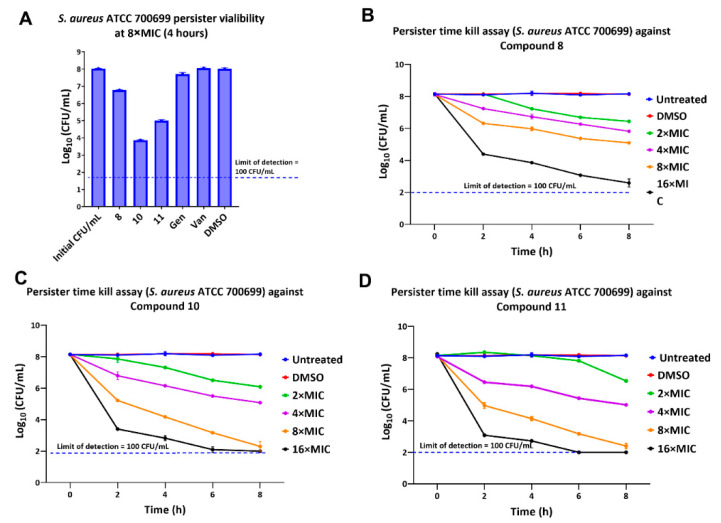
Effect of the lead compounds on the *S. aureus* ATCC 700699 persisters viability. (**A**) Effect of compounds (**8**, **10**, and **11**), positive controls (gentamicin and vancomycin), and DMSO on persisters. Effect of different concentrations of lead compounds **8** (**B**), **10** (**C**) and **11** (**D**) at different time points on *S. aureus* persisters. Each data point in all graphs represents the mean of viable colony counts performed in triplicates and error bars represent the standard deviation.

**Figure 7 antibiotics-11-00939-f007:**
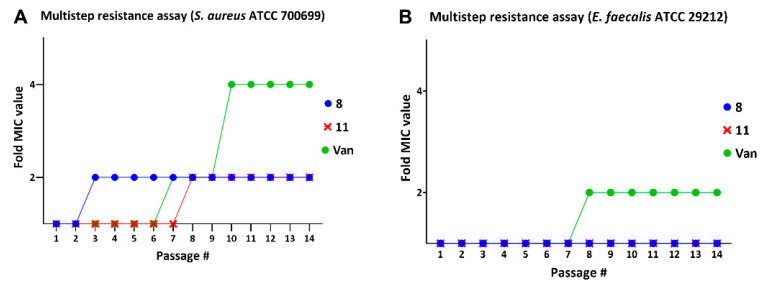
Multistep resistance assay of potent compounds against (**A**) *S. aureus* ATCC 700699 and (**B**) *E. faecalis* ATCC 29212. Vancomycin (van) is a positive control and resistance studies were done for 14 days.

**Figure 8 antibiotics-11-00939-f008:**
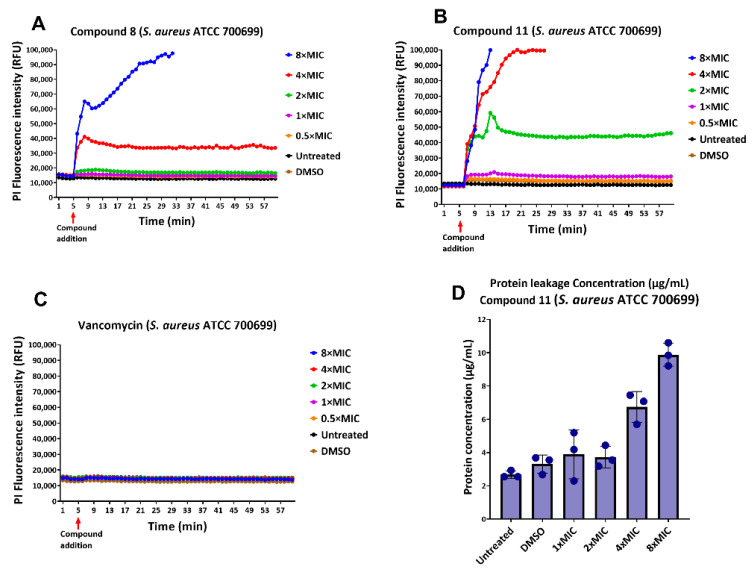
Kinetic fluorescence measurements to detect membrane permeabilization of *S. aureus* ATCC 700699 caused by compounds **8** (**A**), **11** (**B**) and Vancomycin (**C**) at various concentrations using propidium iodide. Each data point in the line graphs represents mean value of PI fluorescence performed in triplicates. Protein leakage concentration against compound **11** is shown in bar graph (**D**). Each value in the bar graph represents the mean of protein concentration measured in triplicates and error bars represent standard deviation.

**Figure 9 antibiotics-11-00939-f009:**
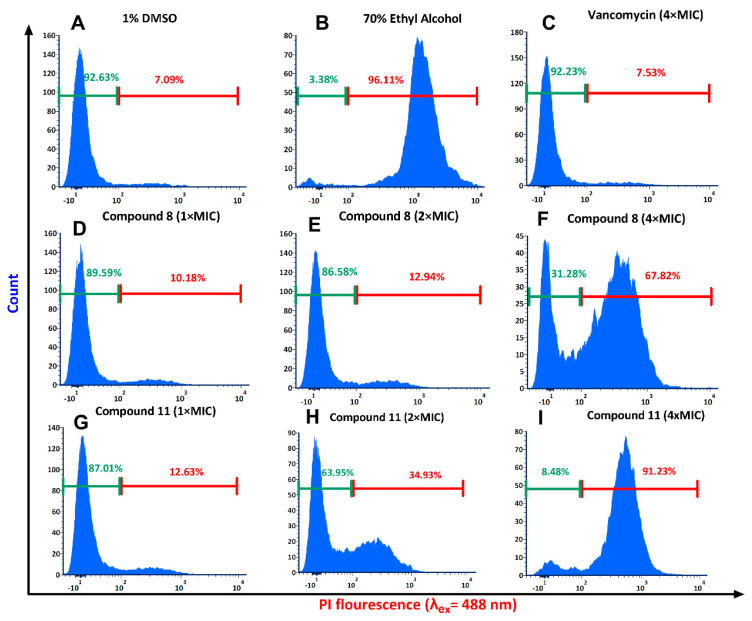
Flow cytometry analysis of bacteria treated with potent compounds. The membrane permeability of *S. aureus* ATCC 700699 treated by compounds at various concentrations and controls was measured by an increase of fluorescent intensity of propidium iodide at 4 °C for 30 min. (**A**) 1% DMSO; (**B**) 70% ethanol; (**C**) Vancomycin at 4 × MIC; (**D**) Compound **8** at 1 × MIC; (**E**) Compound **8** at 2 × MIC; (**F**) Compound **8** at 4 × MIC; (**G**) Compound **11** at 1 × MIC; (**H**) Compound **11** at 2 × MIC ; (**I**) Compound **11** at 4 × MIC**3**.

**Table 1 antibiotics-11-00939-t001:** MIC values of the lead compounds (µg/mL): antibiotic susceptible *S. aureus* ATCC 25923 (Sa23), antibiotic-resistant *S. aureus* ATCC 700699 (Sa99), *S. aureus* BAA-2312 (Sa12), *S. aureus* ATCC 33592 (Sa92), *S. aureus* ATCC 33591 (Sa91), *S. aureus* Newman (SaN), *S. aureus* USA300 (Sa00), S. *aureus* UAMS-1 (Sa1), vancomycin-resistant *E. faecium* ATCC 700221 (Ef21), antibiotic susceptible *E. faecalis* ATCC 29212 (Ef12), *E. faecalis* ATCC 51299 (Ef99), *Bacillus subtilis* ATCC 6623 (Bs), *S. epidermidis* ATCC 700296 (Se). Vancomycin (Van), and Daptomycin (Dap) were used as positive control. DMSO (2.5%) and growth media were used as negative controls.

Comp	Sa23	Sa99	Sa12	Sa92	Sa91	SaN	Sa00	Sa1	Ef12	Ef21	Ef99	Bs	Se
**1**	25	25	25	50	25	>50	>50	>50	>50	50	>50	25	50
**2**	12.5	12.5	12.5	25	12.5	12.5	12.5	12.5	12.5	50	25	6.25	25
**3**	12.5	12.5	12.5	25	12.5	12.5	12.5	12.5	>50	50	25	6.25	25
**4**	12.5	12.5	12.5	12.5	12.5	6.25	12.5	12.5	12.5	25	25	6.25	25
**5**	25	25	50	50	25	>50	>50	>50	50	>50	25	12.5	50
**6**	6.25	3.12	6.25	6.25	3.12	3.12	6.25	6.25	6.25	12.5	6.25	3.12	12.5
**7**	3.12	6.25	6.25	12.5	6.25	3.12	6.25	6.25	6.25	12.5	6.25	3.12	12.5
**8**	1.56	1.56	0.78	3.12	0.78	1.56	1.56	1.56	3.12	3.12	3.12	0.78	6.25
**9**	6.25	12.5	12.5	12.5	6.25	3.12	12.5	12.5	6.25	25	25	3.12	25
**10**	3.12	3.12	3.12	3.12	3.12	3.12	3.12	3.12	3.12	6.25	6.25	1.56	6.25
**11**	3.12	3.12	3.12	3.12	3.12	3.12	3.12	3.12	3.12	6.25	6.25	1.56	6.25
**Van**	0.78	3.12	0.78	1.56	1.56	0.78	0.78	0.78	3.12	>50	>50	0.19	3.12
**Dap**	1.56	6.25	0.78	3.12	6.25	3.12	3.12	3.12	12.5	12.5	12.5	0.78	0.78

**Table 2 antibiotics-11-00939-t002:** MBC values of the potent compounds against susceptible bacteria.

Comp	Bs	Sa12	Sa23	Sa99	Sa91	Sa92	Se
**6**	12.5	50	50	50	12.5	50	50
**7**	12.5	50	25	25	12.5	50	50
**8**	3.12	3.12	12.5	6.25	3.12	12.5	12.5
**10**	6.25	12.5	25	12.5	12.5	25	25
**11**	12.5	25	12.5	12.5	12.5	25	25
**Van**	1.56	1.56	12.5	6.25	6.25	6.25	3.12

**Table 3 antibiotics-11-00939-t003:** MIC and MBEC values of the three lead compounds against Sa23, Sa00 and EFs12 determined by Calgary Biofilm methods.

Comps	Sa23		Sa00		Efs12	
	MIC	MBEC	MIC	MBEC	MIC	MBEC
**8**	1.56	3.12	1.56	12.5	3.12	6.25
**10**	3.12	6.25	3.12	50	3.12	12.5
**11**	3.12	6.25	3.12	25	3.12	6.25
**Van**	0.78	>50	0.78	>50	3.12	>50

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
