# Peer review of "Development of 4-[4-(Anilinomethyl)-3-phenyl-pyrazol-1-yl] Benzoic Acid Derivatives as Potent Anti-Staphylococci and Anti-Enterococci Agents"

_antibiotics, 2022, doi:10.3390/antibiotics11070939_

Round 1

Reviewer 1 Report

In this paper, eleven 4-[4-(Anilinomethyl)-3-phenyl-pyrazol-1-2yl]benzoic acid derivatives have been identified as potent anti-Gram-positive bacterial agents. Meanwhile, the plausible mode of action of the lead compounds was studied and the results showed that they can enhance the permeabilization of the cell membrane. This work are very meaningful. It should be published after minor revision.

1. In line 78,author got the conclude that “Overall, disubstituted compounds showed better activity than that of the mono-substituted compounds”. How explain the compound 5 is weaker than compounds 2, 3, 4.

2. In line 418, author should indicate which two compounds are very potent biofilm inhibitors at 2×MIC doses.

3. In your introduction author should briefly describe why you chose the 11 compounds.

4. The structure of Figure 1 should be changed to the general formula.

Author Response

We thank the reviewer for valuable time and useful comments to improve the manuscript. 

  1. In line 78,author got the conclude that “Overall, disubstituted compounds showed better activity than that of the mono-substituted compounds”. How explain the compound 5 is weaker than compounds 2, 3, 4.

We added the following statement “except the 4-fluoro-3-methyl substituted derivative (5).

  1. In line 418, author should indicate which two compounds are very potent biofilm inhibitors at 2×MIC doses.

We added the “(8 and 11)” in the abstract.

  1. In your introduction author should briefly describewhy you chose the 11 compounds.

We added “(MIC ≤ 32 μg/mL)” in the 1st para of the results and discussion section.

  1. The structure of Figure 1 should be changed to the general formula.

We have shown the structure of each compound for clarity.

Reviewer 2 Report

The present work deals with evidently actual problem: the search for novel antibacterial agents effective against antibiotic-resistant strains.

Though the featured compounds have been previously reported by the authors as antibacterial agents, the present work is focused on the investigation of activity of the selected pyrazolylbenzoic acids against antibiotic-resistant S. aureus and E. faecalis strains. The antibacterial properties of the compounds were estimated via a broad set of assays, including those on bacterial films, to give promising results; also the plausible mode of action was studied for the first time for these compounds.  Therefore, the novelty and significance of the work seem quite satisfactory for the publication.

Surprisingly, the authors do not comment their work in EJMC2021 https://doi.org/10.1016/j.ejmech.2021.113402  , ref.30, dealing with similar compounds, including those with significant better results. I found it sort of misleading. That paper must be cited more evidently, than just a source of the methods used, and the results obtained in the present study may become the continuation of the SAR analysis from the previous work.

Besides, the following minor corrections should be made.

1)     According to IUPAC rules, the name of the parental structure is 4-(3-phenyl-4-((phenylamino)methyl)-1H-pyrazol-1-yl)benzoic acid, not  4-[4-(anilinomethyl)-3-phenyl-pyrazol-1-2 yl]benzoic acid.

2)     Page 1, line 34.  Reference (https://www.frontiersin.org/articles/10.3389/fmicb.2021.737635/full) should be properly cited and inserted into the list of references.

3)     Formatting of the article should be revised. See legends to Figures 3,5,9 and beginning of the part 2.7.

4)     The mentioned characteristics of “lead compounds”, which have been obtained earlier, should be given (antimicrobial activity, toxicity, lipophilicity).

5)     I would carefully use the term “lead compounds” dealing with a set of molecules, though active, but not characterized with clear molecular target and ADME parameters and not implying further structure optimization. Probably, “hit compounds” would be more correct.

6)     References for the synthesis of compounds 1-11 should be given in the “Materials and Methods” section

Author Response

Surprisingly, the authors do not comment their work in EJMC2021 https://doi.org/10.1016/j.ejmech.2021.113402  , ref.30, dealing with similar compounds, including those with significant better results. I found it sort of misleading. That paper must be cited more evidently, than just a source of the methods used, and the results obtained in the present study may become the continuation of the SAR analysis from the previous work.

We added this reference with our other references in the results and discussion section.

Besides, the following minor corrections should be made.

  • According to IUPAC rules, the name of the parental structure is 4-(3-phenyl-4-((phenylamino)methyl)-1H-pyrazol-1-yl)benzoic acid, not  4-[4-(anilinomethyl)-3-phenyl-pyrazol-1-2 yl]benzoic acid.

Drawing tools showing the IUPAC name of the general structure as 4-[4-(anilinomethyl)-3-phenyl-pyrazol-1-yl]benzoic acid.

  • Page 1, line 34.  Reference (https://www.frontiersin.org/articles/10.3389/fmicb.2021.737635/full) should be properly cited and inserted into the list of references.

We corrected this mistake.

  • Formatting of the article should be revised. See legends to Figures 3,5,9 and beginning of the part 2.7.

We formatted the article.

  • The mentioned characteristics of “lead compounds”, which have been obtained earlier, should be given (antimicrobial activity, toxicity, lipophilicity).

We are not sure about this comment.

  • I would carefully use the term “lead compounds” dealing with a set of molecules, though active, but not characterized with clear molecular target and ADME parameters and not implying further structure optimization. Probably, “hit compounds” would be more correct.

We appreciate the reviewer for this comment. We change the ‘lead’ with ‘hit’ in the manuscript.

6)     References for the synthesis of compounds 1-11 should be given in the “Materials and Methods” section

We added a section in the material and methods section for the compounds.